# Effects of Dietary Glucose Oxidase Supplementation on the Performance, Apparent Ileal Amino Acids Digestibility, and Ileal Microbiota of Broiler Chickens

**DOI:** 10.3390/ani11102909

**Published:** 2021-10-08

**Authors:** Yong Meng, Haonan Huo, Yang Zhang, Shiping Bai, Ruisheng Wang, Keying Zhang, Xuemei Ding, Jianping Wang, Qiufeng Zeng, Huanwei Peng, Yue Xuan

**Affiliations:** 1Feed Engineering Research Centre of Sichuan Province, Institute of Animal Nutrition, Sichuan Agricultural University, Chengdu 611130, China; mengyong219@163.com (Y.M.); m18482109245@163.com (H.H.); zkeying@sicau.edu.cn (K.Z.); dingxuemei0306@163.com (X.D.); wangjianping@sicau.edu.cn (J.W.); zqf@sicau.edu.cn (Q.Z.); phw@sicau.edu.cn (H.P.); xuanyuede1007@hotmail.com (Y.X.); 2Mianyang Habio Bioengineering Co., Ltd., Mianyang 610000, China; zhangy0523@126.com; 3Chongqing Academy of Animal Science, Chongqiang 402460, China; short00@163.com

**Keywords:** glucose oxidase, growth performance, amino acid digestibility, ileal microbiota, broiler

## Abstract

**Simple Summary:**

Glucose oxidase was used as a potential additive to improve intestinal health in livestock and poultry industry. This study aimed to investigate the effects of glucose oxidase supplementation on performance, ileal microbiota, ileal short-chain fatty acids profile, and apparent ileal digestibility in grower broilers. Our findings will provide a valuable insight into the possibility of glucose oxidase as an alternative of antibiotic growth promoters in broiler diets.

**Abstract:**

This study aimed to investigate the effects of glucose oxidase (GOD) supplementation on growth performance, apparent ileal digestibility (AID) of nutrients, intestinal morphology, and short-chain fatty acids (SCFAs) and microbiota in the ileum of broilers. Six hundred 1-day-old male broilers were randomly allotted to four groups of 10 replicates each with 15 birds per replicate cage. The four treatments included the basal diet without antibiotics (Control) and the basal diet supplemented with 250, 500, or 1000 U GOD/kg diet (E250, E500 or E1000). The samples of different intestinal segments, ileal mucosa, and ileal digesta were collected on d 42. Dietary GOD supplementation did not affect daily bodyweight gain (DBWG) and the ratio of feed consumption and bodyweight gain (FCR) during d 1-21 (*p* > 0.05); however, the E250 treatment increased DBWG (*p* = 0.03) during d 22–42 as compared to control. Dietary GOD supplementation increased the AIDs of arginine, isoleucine, lysine, methionine, threonine, cysteine, serine, and tyrosine (*p* < 0.05), while no significant difference was observed among the GOD added groups. The E250 treatment increased the villus height of the jejunum and ileum. The concentrations of secreted immunoglobulin A (sIgA) in ileal mucosa and the contents of acetic acid and butyric acid in ileal digesta were higher in the E250 group than in the control (*p* < 0.05), whereas no significant differences among E500, E1000, and control groups. The E250 treatment increased the richness of ileal microbiota, but E500 and E100 treatment did not significantly affect it. Dietary E250 treatment increased the relative abundance of *Firmicutes* phylum and *Lactobacillus* genus, while it decreased the relative abundance of genus *Escherichina-Shigella* (*p* < 0.05). Phylum *Fusobacteria* only colonized in the ileal digesta of E500 treated broilers and E500 and E1000 did not affect the relative abundance of *Firmicutes* phylum and *Lactobacillus* and *Escherichina-Shigella* genera as compared to control. These results suggested that dietary supplementation of 250 U GOD/kg diet improves the growth performance of broilers during d 22–42, which might be associated with the alteration of the intestinal morphology, SCFAs composition, and ileal microbiota composition.

## 1. Introduction

Animal production without antibiotics has received much attention in recent years, because many countries have increased regulation on antibiotic growth promoter use in animal feed production. The diet without antibiotics increases the occurrence of gut-intestinal diseases and decreases the growth performance of broiler chickens [1,2]. Recent studies found that glucose oxidase (GOD) is one potential additive to increase intestinal health and the performance of broilers [3]. Dietary inclusion of lower doses of GOD (40-60 U/kg) improves the bodyweight gain and feed efficiency of weaned piglets [4,5,6]. However, there were inconsistent results in the effect of GOD addition on the growth performance in poultry. Wu et al. found that the dietary addition of low doses of GOD (40–60 U/kg) increased the growth performance during day 1–21, whereas it did not significantly increase the growth performance during day 22–42 in fast growing broilers [7]. Low dose of GOD addition also did not significantly increase the growth performance in slow growing broilers from d 1 to 52, [8] and in ducks from d 10 to 28 [9]. These inconsistent results might be attributed to the differences in the dose of GOD, growth period, broiler strain, and poultry species. Recently, the improvement of GOD production technology has increased the GOD activity per unit weight of the products, which made it possible to add relative high doses of GOD into animal diets. However, little research has been conducted on the effects of dietary high dose of GOD on the growth performance in poultry.

Intestinal microbiota plays an important role in nutrient digestibility in the intestine of broiler chicken [10,11]. Wu et al. found that dietary inclusion of *Lactobacillus* altered the microflora composition in the ileum and cecum of broilers, which increased the digestibility of energy, amino acids, and calcium [12]. Rodjan et al. also reported that the addition of *Bacillus* in the diet increased the digestibility of digest crude fiber in broilers [13]. On the other hand, intestinal microbiota also can alter the production of short-chain fatty acids (SCFAs) from nondigestible nutrients fermentation, which affects the integrity of the epithelial barrier and immunoregulatory functions [14]. Besides, the intestinal microbiota is one of the main defense components in the gut of chicken against intestinal pathogenic bacteria, especially *Clostridium perfringens* [15,16]. Previous studies reported that GOD inhibited the growth of foodborne pathogens, including *Clostridium perfringens*, *Campylobacter jejuni*, and *Salmonella infantis* [17,18]. Its products gluconic acid and hydrogen peroxide also showed antimicrobial capacity against *Paenibacillus larvae* ATCC9545 [19]. Dietary inclusion of GOD increased the relative abundance of cecal beneficial bacterium of starter broilers, such as *Firmicutes* and *Ruminococcaceae* [7] and improved the immune function of ducks against *Escherichia coli* O88 [8]. However, little work has been done about the effects of dietary inclusion of GOD on ileal microbiota in poultry. We hypothesized that the relative high dose of GOD altered the ileal microbiota and nutrients digestibility, resulting in the improvement of grower broilers (d 22–42). Therefore, the aim of this study was to investigate the effects of high dose of GOD addition (over 250 U/g) on the performance, ileal amino acid digestibility, the concentrations of SCFAs, and ileal microbiota in broiler chickens.

## 2. Materials and Methods

All experimental procedures were approved by the Institutional Animal Care and Use Committee of Sichuan Agricultural University (Ethical code: 2018–121) and were performed strictly following the guidelines set by the National Institute of Animal Health. 

### 2.1. Animals and Diets

Six hundred 1-day-old male broiler chicks (Arbor Acres strain) were randomly allotted to 1 of 40 suspended wire cages, so that there were 15 chicks per cage. The 4 dietary treatments included the basal diet (Control) and the basal diet added with 250, 500, or 1000 IU GOD/kg diet (E250, E500, or E1000), which corresponded to the basal diet added with 0, 50, 100, or 200 mg GOD product/kg diet (5000 IU/g GOD product). The GOD was provided by Mianyang Habio Bioengineering Co., Ltd. (Mianyang, China), and the activity of GOD was analyzed before adding into the diets. The experimental units (cages) were distributed randomly to the 4 dietary treatments with 10 replicates per treatment. The basal diets (Table 1) were formulated to meet or exceed the nutrient requirements recommended by the Feeding Standard of Chicken [20]. All diets were subjected to cold pelleting (60 to 65 °C). 

### 2.2. Animal Management

The experiment was carried out on the farm of the Animal Nutrition Institute of Sichuan Agricultural University, China. All birds were reared in suspended cages (100 × 200 × 45 cm) in an environmentally controlled house. The temperature was maintained at 32 °C during the first two days. Then, the ambient temperature was gradually decreased by 1 °C every two days until it reached 22 °C and was maintained at this temperature until the end of the experiment. All birds were housed under a 16-h light and 8-h dark cycle and the feed and water were provided ad libitum. The relative humidity of the house was maintained at 50%.

### 2.3. Growth Performance and Carcass Traits

On d 21 and 42, the chickens and non-consumed feed were weighed for each replicate to determine feed consumption and bodyweight gain and calculate daily bodyweight gain (DBWG), daily feed intake (DFI), and the ratio of feed consumption: bodyweight gain during d 1–21, d 22–42, and d 1–42. The number of chickens was checked every day, and the number and weight of dead and eliminated chickens was recorded. The mortality was calculated during d 1–42. On d 42, one bird per replicate was selected, whose bodyweight was close to the average bodyweight of the replicate. All selected birds were euthanized with CO_2_. After exsanguination, the head, feather, feet, digestive organs, abdominal fat (comprising fat surrounding the abdominal organs), and internal organs (heart, liver, spleen, bursa, and lung) of the birds were removed. Then, the carcass was spraying-washed, immersion-chilled at 2 °C for 30 min, and drained effectively for 5 min. The carcass yield was calculated as a percentage of the live bodyweight. The leg muscle, breast muscle, and abdominal fat were weighted and presented as a percentage of the live bodyweight. 

### 2.4. Sample Collection

On d 42, three birds per replicate were randomly selected and were euthanized with CO_2_. After exsanguination, the mid segments (about 1 cm) of duodenum, jejunum, and ileum in one bird per replicate were collected and fixed in 10% of paraformaldehyde solution for the intestinal morphology analyses. The remaining duodenal, jejunal, or ileal segments were used to collect the mucosal samples for secreted immunoglobulin A (sIgA) analysis. The digesta in the duodenum, jejunum, and ileum was collected to determine pH value. For another two birds, the content of ileum was collected aseptically, mixed to one sample, and stored at −20 °C until the SCFAs and ileal microbiota analyses. 

### 2.5. Apparent Ileal Nutrient Digestibility Assay

During d 36–42, all birds were fed the diets added with 0.3% Cr_2_O_3_ as an inert marker. We have one week of preliminary feeding to let the broilers adapt to the diet contained the Cr2O3. On d 42, 4 birds per replicate were randomly selected and euthanized by intravenous injection of sodium pentobarbitone (80 mg/kg BW; Sigma-Aldrich, Shanghai, China). The abdominal cavities of the birds were then opened immediately and the ileum segment was excised from Meckel’s diverticulum to 2 cm proximal to the ileo-cecal junction. The digesta in the distal half of the ileum was collected by gently flushing with distilled water into a plastic container. The ileal contents from the birds within each replicate were pooled into one sample. All digesta samples were frozen immediately after collection at −20 °C until analysis.

### 2.6. Chemical Analyses

All digesta samples were freeze-dried at −50 °C (FDU-2110, EYELA, Tokyo, Japan). Before analyses, the diets and dried digesta samples were ground by an analytic mill processing to pass through a 0.18 mm screen. The diets and digesta samples were analyzed for gross energy, crude protein, and amino acids on a dry matter (DM) basis. The DM was determined by drying at 105 °C for 24 h (method 934.01; AOAC, 2005). The gross energy was determined using a bomb calorimeter (Parr Instrument Co., Moline, IL, USA). The concentration of crude protein was measured according to the method of AOAC (method 990.3; AOAC, 2005). The concentrations of amino acids were determined using an amino acid analyzer (L-8900, Hitachi, Tokyo, Japan). In brief, about 0.2 g sample was hydrolyzed by 6 N HCl containing 6 g/L phenol for 24 h at 110 °C in a glass tube sealed under vacuum. For the cysteine and methionine analyses, the samples were oxidized by performic acid for 16 h at 0 °C, and then the reaction was terminated by adding hydrobromic acid before acid hydrolysis. The concentration of chromium was determined by digesting the samples in concentrated nitric acid and perchloric acid, and the absorption was measured using a spectrophotometer at 440 nm (Beckman DU-800; Beckman Coulter, Inc., Brea, CA, USA). All analyses were performed in duplicate. 

### 2.7. Intestinal Morphology, Secreted Immunoglobulin A, and pH of Intestinal Digesta

The paraformaldehyde-fixed duodenal, jejunal, and ileal segments were dehydrated in graded alcohol series and then embedded in paraffin. After that, the paraffin blocks were sectioned to 5 μm thickness by a microtome, and then were put on a glass slide and stained with hematoxylin and eosin for microscopic evaluation. Morphological evaluation included villi height (VH; the distance from the apex of the villus to the junction of the villus and crypt) and crypt depth (CD; the distance from the junction to the basement membrane of the epithelial cells at the bottom of the crypt). Then, the ratio between villus height and crypt depth (VH/CD) was calculated for each treatment group. The concentration of secreted immunoglobulin A (sIgA) in the duodenal, jejunal, or ileal mucosa samples was determined by ELISA kits (Nanjing Jiancheng Biological Engineering Research Institute, Nanjing, China). The pH of duodenal digesta, jejunal digesta, and ileal digesta were measured on fresh samples as described by Yang et al. [21]. In brief, one gram of fresh digesta was diluted with 9 mL of deionized water and then the pH of digesta was determined using the pH meter (HI2210, Hanna Instruments, Inc., Woonsocket, RI, USA).

### 2.8. Short Chain Fatty Acids (SCFAs) Analyses

Concentrations of acetic, propionic, butyric, valeric, and iso-valeric acids in ileal digesta were determined by gas chromatography as described by Bassanini et al. [22]. All analyses were performed using a gas chromatograph (GC-CP3800, Varian, Walnut Greek, CA, USA) equipped with a flame ionization detector and a chromatographic column (30 m × 0.245 mm inner diameter, 0.25 μm film thickness; Agilent, Santa Clara, CA, USA). Briefly, approximately 0.7 g of ileal digesta was suspended in 1.5 mL of distilled water. Then, the mixed solution was centrifuged for 15 min at 15,000× *g*. One milliliter of the homogeneous supernatant was transferred into a conical polypropylene micro sample tube (Eppendorf, 2 mL), and then was added with 0.2 mL of metaphosphoric acid (25%, *w*/*v*) and 23.3 μL of crotonic acid (210 mmol/L). Ten microliters of a solution of 150 mmol/L of 2-ethylbutyric acid in formic acid was added as the internal standard. After that, the mixed solution was cultured for 30 min at 4 °C and then centrifuged for 10 min at 15,000× *g*. The 0.3 mL of supernatant was transferred into a now Eppendorf tube and then 0.9 mL of formic acid was added. After centrifugation (5 min at 10,000× *g*), one microliter of supernatant was injected into the gas chromatograph for analysis. Results are expressed as nmol/g or pmol/g of wet weight of digesta.

### 2.9. DNA Extraction, PCR Amplification, and Sequencing

The DNA was extracted from the six of ten ileal content samples using MiniBEST Bateria Genomic DNA Extraction Kit (TaKaRa Bio Inc., Dalian, China). The extracted DNA was purified by Zymobiomics DNA Microprep kit (Cat# D4301; Zymo Research, Irvine, CA, USA). Then, the DNA concentration was measured by Tecan infinite^®^ 200 (Männedorf, Switzerland) and DNA quality was evaluated using gel electrophoresis. The V4 variable region of the 16S rRNA gene was amplified from the extracted DNA using the barcoded primers 5′-GTGYCAGCMGCCGCGGTAA-3′ (forward) 55-GGACTACHVGGGTWTCTAAT-3AT-3′ (reverse). PCR reaction was performed using a volume of 50 μL premix which contained 0.5 μL KOD-Plus-Neo polymerase (1U/50 μL; TOYOBO Co., Ltd., Osaka, Japan), 5 μL 10 × PCR Buffer, 1.5 μL forward primer (10 μM), 1.5 μL reverse primer (10 μM), 2 μL sample DNA (20 ng/μL), and 39.5 μL H_2_O. The PCR amplification program included 94 °C for 1 min, then 30 cycles of 95 °C for 20 sec, 54 °C for 30 sec, 72 °C for 30 sec, and final extension at 72 °C for 5 min in an Applied Biosystems^®^ PCR System 9700 thermocycler (Thermo Fisher Scientific, Foster City, CA, USA). The PRC product was run on a 2% agarose gel, and the bands of desired size were collected and purified using Zymoclean gel recovery kit (D4008; Zymo Research, Irvine, CA, USA). The concentration of purified DNA was measured by Qubit^®^ 2.0 fluorometer (Thermo Fisher Scientific, Carlsbad, CA, USA). DNA sequencing libraries were constructed using TruSeq DNA PCR-Free Sample Prep kit (Illumina Inc., San Diego, CA, USA). The library was sequenced on the Illumina MiSeq platform using Hiseq Rapid SBS kit v2 (Illumina Inc., San Diego, CA, USA).

### 2.10. Sequence Data Process and Operational Taxonomy Units (OTUs) Clustering

The paired-end reads were assembled using FLASH software with a maximum mismatch rate of 20% [23]. The sequencing data was further denoised by the QIIME software [24], and all sequences showed (1) no mismatch to the barcode, (2) 16S rRNA gene primer at sequencing end, (3) more than 200 nucleotides in length, and (4) no more than two undermined bases. The sequences were compared with the reference database (Silva database, https://www.arbsilva.de/) [25] using UCHIME algorithm [26] to detect and remove chimeric sequences [27]. The remaining sequences were then clustered to generate operational taxonomic units (OTUs) using Usearch software (v 7.1) with UPARSE algorithm with a similarity cutoff of 97% [28]. The representative sequence was annotated using the UCLUST method and Siliva reference library [29].

### 2.11. Bioinformatics Analyses

All microbiome data analyses were performed in R 3.6.0 software. Observed OTUs and α-diversity indices, such as Chao1, ACE, Simpson, and Shannon indices were calculated using the vegen package. A comparison of those indices was performed using the ANOVA on ranks with the Kruskal–Wallis rank nonparametric test. The weighted and unweighted Unifrac distances were calculated using the gunifrac package. Principal Coordinate Analysis (PCoA) utilizing the weighted and unweighted UniFrac distances were calculated using ape package. Additionally, permutational MANOVA was carried out using vegan package measure effect size and significance on β-diversity. Ileal bacterial taxonomic comparison at phylum and genus level were performed among different treatments using Kruskal–Wallis rank sum test.

### 2.12. Calculations and Statistical Analysis

Apparent ileal digestibility (AID) of energy, crude protein, and amino acids were calculated using the following equation:$$AID\;\left(\%\right)=\left[1-(Cr_{diet}/Cr_{digesta}\right)\times \left(N_{digesta}/N_{diet}\right)]\times 100,$$
where *AID* is the apparent ileal digestibility of nutrient or energy (%), *Cr_diet_* is the concentration of chromium in the diet, *Cr_digesta_* is the concentration of chromium in the digesta, *N_digesta_* is the concentration of nutrients in the digesta, and *N_diet_* is the concentration of nutrients in the diet.

The experimental data was statistically analyzed by using the GLM procedure of SAS 9.2. A replicate cage was used as an experimental unit for analyzing the data of performance, nutrients digestibility, SCFAs, and ileal microbiota, whereas each bird selected was used as the statistical unit for analyzing the data of intestinal morphology, sIgA and pH of intestinal digesta. The data were expressed by least square means and pooled standard errors. The means were separated using Tukey’s test and *p* < 0.05 was considered as a significant difference.

## 3. Results

### 3.1. Growth Performance and Carcass Traits

The effect of dietary inclusion of GOD on the performance of broilers is shown in Table 2. Compared with the control, the GOD supplementation did not significantly affect the BW of broilers at 21 days of age, and the DBWG, DFI, and FCR during d 1-21 (*p* > 0.15). However, during d 22–42, the DBWG was higher (*p* = 0.03) in E250 treatment than in the control, and no significant difference was observed among the control, E500, and E1000 groups. During the whole period (d 1–42), dietary GOD supplementation did not significantly influence the DBWG, DFI, FCR, and mortality (*p* > 0.23). About the carcass characteristic of broilers, we found that dietary inclusion of GOD did not significantly affect the carcass yield, eviscerated carcass yield, thigh yield, breast yield, and abdominal fat weight (Appendix A).

### 3.2. Apparent Ileal Nutrient Digestibility

The effects of dietary inclusion of GOD on the apparent ileal digestibility (AID) of nutrients are shown in Table 3. Dietary GOD supplementation did not significantly affect the AIDs of dry matter, energy, and crude protein as compared to the control (*p* > 0.07). However, it increased the AIDs of arginine, isoleucine, lysine, methionine, threonine, cysteine, serine, and tyrosine (*p* < 0.05) as compared to the control regardless of added GOD level, and no significant difference was observed among the different GOD added treatments. Dietary GOD addition did not significantly affect the AIDs of histidine, leucine, phenylalanine, valine, alanine, aspartic acid, glutamic acid, glycine, and proline.

### 3.3. Intestinal Morphology and Secreted Immunoglobulin A (sIgA)

The effects of dietary inclusion of GOD on intestinal morphological parameters and sIgA concentration in the intestinal mucosa are shown in Table 4. No significant difference was observed for the VH, CD, and the ratio of VH: CD in the duodenum of broilers at 42 days of age among all treatments (*p* > 0.37). The E250 treatment increased the VH (*p* = 0.0042), but had no significant effect on the CD and VH: CD ratio in the jejunum of broiler as compared to other treatments. No significant differences in the CH and VH: CD ratio of jejunum were observed among different treatments. In the ileum, the E250 and E500 treatments increased the VH (*p* < 0.001), but they did not affect the CD and VH: CD ratio as compared to the control. No significant difference was observed between the control and E1000 groups for the VH, CD, and VH: CD ratio in the ileum of broilers. The E250 treatment increased the concentration of sIgA in the ileum when compared to the control and E500 groups. However, dietary GOD addition did not significantly affect the concentration of sIgA in the duodenal and jejunal mucosa of broilers.

### 3.4. pH Value of Intestinal Contents

The effect of dietary inclusion of GOD on the pH of intestinal digesta is shown in Table 5. No significant difference was observed in the pH of duodenal digesta among all treatments (*p* = 0.40). However, the E500 treatment decreased the pH of jejunal digesta as compared with the other groups (*p* < 0.05). The pH of ileal digesta was lower (*p* < 0.04) in the E250 and E1000 treatments than that in the control and E500 groups.

### 3.5. Short Chain Fatty Acids Concentration

The effect of dietary inclusion of GOD on the concentrations of SCFAs in the ileal digesta is shown in Table 6. The E250 treatment increased the concentrations of acetic acid, butyric acid, and total SCFAs as compared to other groups (*p* < 0.05), while no significant difference was observed among the E500, E1000 and control treatments. Dietary supplementation of GOD had no significant effect on the concentrations of propionic acid, isovaleric acid and valeric acid (*p* > 0.10).

### 3.6. Biodiversity of the Ileal Microbiota

Illumine sequencing displayed that a total of 794,875 bacterial 16S rRNA high-quality reads were obtained from 24 ileal DNA samples. The sequences were further clustered into 2,105 operational taxonomic units (OTUs) using a 97% similarity cut off. The Venn diagram showed that 314 of 2105 OTUs were shared among different treatments, representing 96.51% the richness of ileal microbiota. Notably, the 553, 27, 390, and 85 OTUs were unique for the control, E250, E500, and E1000 groups, respectively (Appendix A). The alpha diversity indices for each group are presented in Table 7. The E250 treatments significantly decreased the number of observed species as compared to the control and E500 groups (*p* < 0.05). The Chao and ACE indices were lower in E250 treatment than in control and E500 treatments (*p* < 0.05), while no significant difference was observed between E1000 and other treatments (*p* > 0.10). Dietary supplementation of GOD had no significant effect on the Simpson and Shannon indices (*p* > 0.36).

The similarity between bacterial communities present in each group is represented by principal coordinate analysis (PCoA) in Figure 1. In the PCoA plot based on the unweighted UniFrac distance, the E250 significantly changed the ileal microbial structure (PERMANOVA; *r* = 0.18, *p* = 0.049), whereas the E500 and E1000 did not significantly affect it (*p* > 0.07) as compared to the control (Figure 1, Appendix A). The ileal microbiota structure of the E500 group had significantly diverged from that of the E250 and E1000 (PERMANOVA; *r* = 0.17, *p* = 0.012 and *r* = 0.15, *p* = 0.043, respectively). However, the PCoA plot based on weighted UniFrac showed that a significant difference was observed in the ileal microbiota structure between E250 and E500 treatments (PERMANOVA; *r* = 0.28, *p* = 0.033), and there was no significant difference between the control and E250, control and E1000, and E500 and E1000 groups (Appendix A).

### 3.7. Composition of Ileal Microbiota

Relative abundance of the main phyla and genera (top 10) are presented in Figure 2. The *Firmicutes*, *Proteobacteria*, and *Bacteroidetes* phyla accounted for more than 98% of all sequences. The E250 treatment significantly increased the relative abundance of *Firmicutes*, but decreased *Proteobacteria*, *Acidobacteria*, *Spirochaetes*, and *Euryarchaeota* at the phylum level as compared to the control (*p* < 0.05). However, no significant difference was observed in the relative abundance of the *Firmicutes*, *Proteobacteria*, *Bacteroidetes*, *Acidobacteria*, and *Spirochaetes* phyla among the control, E500 and E1000 groups (*p* > 0.11). The *Fusobacteria* phylum colonized in the ileal digesta of E500 treated birds, but not in the other treatment broilers. At the genus level, the *Lactobacillus*, *Escherichia-Shigella*, *Enterococcus*, and *Candidatus Arthromitus* genera accounted for 87.1–97.9% of all sequences. The E250 treatment significantly increased the relative abundance of *Lactobacillus*, but decreased the abundance of *Escherichia-Shigella*, *Candidatus Arthromitus*, *Ruminococcaceae* UCG-002, and *Lachnospiraceae* NK4A136 group as compared to the control (*p* < 0.05). However, no significant difference was observed in the relative abundance of the *Lactobacillus*, *Escherichia-Shigella*, Candidatus Arthromitus, *Bacteroides*, *Ruminococcaceae* UCG-002, and *Lachnospiraceae* NK4A136 group genera among the control, E500, and E1000 groups (*p* > 0.10).

## 4. Discussion

The GOD catalyzes the process of β-d-glucose oxidation into gluconic acid using atomic oxygen as the electron acceptor with the synchronous generation of hydrogen peroxide, which improves the intestinal environment of animals [3]. In the present study, dietary supplementation of 250 U GOD/kg diet increased DBWG in broilers during the grower period (d 22–42). Similarly, Zhao et al. found that the addition of 200 U/kg GOD decreased the FCR in the slow growing broiler chicken during d 42–77 [30]. However, several studies reported that the lower dose of GOD (40–60 and 70 U GOD/kg diet, respectively) had no effect on the performance in the fast-growth broilers during d 22–42 [7,31] and in the slow growing broilers during d 1–52 [8]. The disparities of these results at least partly be attributed to the differences in the dose of added GOD in the diet and the broilers stains. In contrast, our results demonstrated that the GOD addition did not improve the performance of broilers during day 1 to 21, which was inconsistent with the previous finding that the addition of 40–60 U GOD/kg diet improved the FCR of starter broilers (day 1–21) [7,31]. One explanation of this discrepancy was that high dose of glucose oxidase oxidized too much glucose [3], which would impair intestinal development in broilers [32]. Our data also found that the high dose of GOD (more than 250 U/kg) did not further improve the performance of broilers during d 22–42. It was possible that the 250 U GOD/kg diet could meet the requirement of oxidation of intestinal glucose.

In order to explore the mechanism of GOD improving the performance of grower broilers (d 22–42), we measured the nutrients digestibility and intestinal morphology on d 42. Our results demonstrated that dietary GOD addition increased the AIDs of amino acids, which might cause the improvement of the broilers’ performance. Similarly, dietary inclusion of 60 U GOD/kg increased apparent digestibility of Ca, P, and CP in broilers on day 42 [7,31]. Our results of the increases in the jejunal and ileal VH of E250 treated broilers indicated that the appropriate dosage of GOD addition improved intestinal absorption function in broilers. In agreement, previous studies found that dietary GOD addition protected the integrity of the intestinal mucosa and tight junction and increased nutrients absorption in the intestine of broilers [8,33]. The increase in ileal butyric acid and the decrease in the pH of ileal digesta presented here would increase amino acid transport in the intestine of animals [34,35]. Similarly, Wu et al. found that the decrease in the pH of intestinal content caused by dietary GOD addition activated intestinal digestive enzymes, such as pepsin and chymotrypsin, promoted the digestion and absorption of nutrients, and improved nutrient metabolism rate of broilers [7]. Additionally, glucose and some amino acids shared the same transporters [36,37,38]. Our result of decrease in intestinal glucose caused by GOD addition might cause the increase in the uptake of some amino acids in the intestine of broilers.

Ileal microbiota can regulate intestinal morphology and the production of SCFAs from nondigestible nutrients fermentation in the hindgut of animals [14]. Our results confirmed that the most abundant phyla are *Firmicutes*, *Proteobacteria*, and *Bacteroidetes* in the ileum of chicken [13] and the most abundant genus is *Lactobacillus*, followed by two minor genera, *Esherichia coli-Shigella* and *Enterococcus* [39]. The members of the *Firmicutes* mainly produce butyrate in the human gut, while *Bacteroi**detes* mostly produce propionate [40]. Thus, the present result of the increase in ileal butyrate concentration by dietary 250 U/kg GOD addition might be due to the increase in the abundance of *Firmicutes*. The *Lactobacillus* is the main lactic acid-producing bacteria [41]. Our study found that dietary 250 U/kg GOD also increased the concentrations of lactic acid and total SCFAs, which at least partly attributed to the increased abundance of *Lactobacillus* genus in the ileal digesta of broilers. Besides, the ileal microbiota is one of the main defense components in the gut of chicken against intestinal pathogenic bacteria [15,16]. In this study, dietary E250 supplementation decreased the richness of the ileal microbiota of broilers evaluated by the Chao and ACE indexes of microbiota. Our result of the decrease in the number of different species (OTUs) also indicated that dietary inclusion of E250 decreased the diversity of ileal microbiota in broilers. However, dietary E250 supplementation increased the relative abundance of *Lactobacillus* genus. Previous studies found that the increase in *Lactobacillus* abundance accompanied by the increases in the concentrations of lactic acid and total SCFAs could inhibit the replication of some Gram-negative pathogens, including *E. coli* [42,43]. Our data of the decrease in *Esherichia coli-Shigella* abundance caused by E250 supplementation also supported this idea. Additionally, the previous researches suggested that the increase in butyric acid could increase the absorptive surface of the small intestine by stimulation of epithelial cell proliferation and differentiation, leading to better nutrient utilization [31,44]. The present results of the decrease in the pH of ileal digesta and the increase in the concentrations of acetic acid and butyric acid indicated that the 250 U GOD/kg addition improves intestinal acid-base balance environment, resulting in increasing the nutrients absorption in the small intestine of broilers.

*Candidatus Arthromitus* is characterized by their attachment to the intestinal epithelium and its important role in modulating the host immune function [45]. Previous studies found that a decrease of *Candidatus Arthromitus* was associated with increased concentration of intestinal IgA in the ileal mucosa of chicken [46,47,48]. Thus, our results of the decreased abundance of *Candidatus Arthromitus* accompanied by the increase in ileal sIgA suggested that the 250 mg/kg GOD increased ileal immune function of broilers against pathogens infection. However, we also found that the *Fusobacteria* appeared in the ileal digesta of broilers fed with E500 treated diet, but not in the other groups. The increase of *Fusobacteria*, a commensal-turned pathogen, is linked with an excessive inflammatory response in the gut of animals [49]. Our finding of the presence of *Fusobacteria* only in E500 group indicated that the E500 addition had the potential harm to the intestinal morphology and function in broilers. We also found that a high dose of GOD (E500 and E1000) did not significantly affect the ileal microbiota structure when compared to the control, not as well as the E250 treatment. Bedford and Apajalahti [50] found that the microflora population was dependent upon the fermentable substrates from complex plant polymeric compounds such as cellulose, xylan and proteins to the simplest monomeric sugars, amino acids, and fatty acids. Our data of no change in the ADI of total amino acids among GOD added treatments suggested that the differences in the ileal microbiota structure between E250 and E500 or E1000 might not be attributed to the differences in the digestibility of amino acids. Wu et al. demonstrated that the GOD addition increased the activity of ileal amylase that could digest the starch into maltose, monosaccharide, and glucose [7]. Unfortunately, we did not determine the composition of ileal digesta, especially the fiber, resistant starch, and monosaccharide. Thus, further research on the effects of GOD on the digestibility of carbohydrates is necessary.

## 5. Conclusions

Dietary supplementation of 250 U GOD/kg increased the body weight gain in broilers during day 22–42. Dietary supplementation of 250–1000 U GOD/kg increased the AIDs of arginine, isoleucine, lysine, methionine, threonine, cysteine, serine, and tyrosine in grower broilers (day 22–42). Dietary supplementation of 250 U GOD/kg improved the intestinal morphology of broilers, and increased ileal immune function and the concentrations of SCFAs in ileal digesta; however, dietary addition of 500 or 1000 U GOD/kg did not significantly affect these indicators. Simultaneously, dietary supplementation of 250 U GOD/kg treatment decreased the richness of ileal microbiota and altered the ileal microbiota structure in broilers, while dietary addition of 500 or 1000 U GOD/kg did not alter these indicators compared with the control.

## Figures and Tables

**Figure 1 animals-11-02909-f001:**
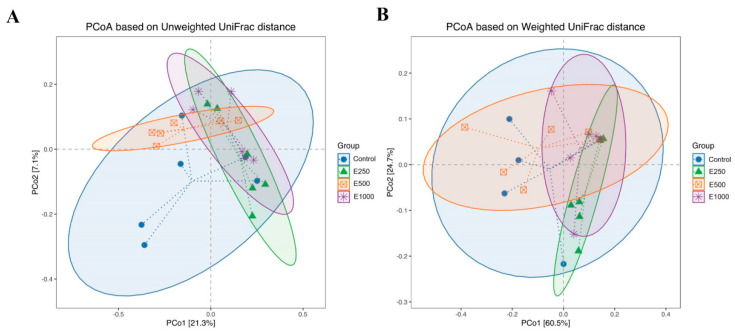
Principal coordinates analysis (PCoA) plot of ileal microbiota based on unweighted (**A**) and weighted (**B**) UniFrac metrics. Ileal digesta samples (*n* = 6) were collected in the broilers after feeding the basal diet supplemented with 0 (Control), 250 (E250), 500 (E500), or 1000 (E1000) U glucose oxidase/kg diet for 42 days.

**Figure 2 animals-11-02909-f002:**
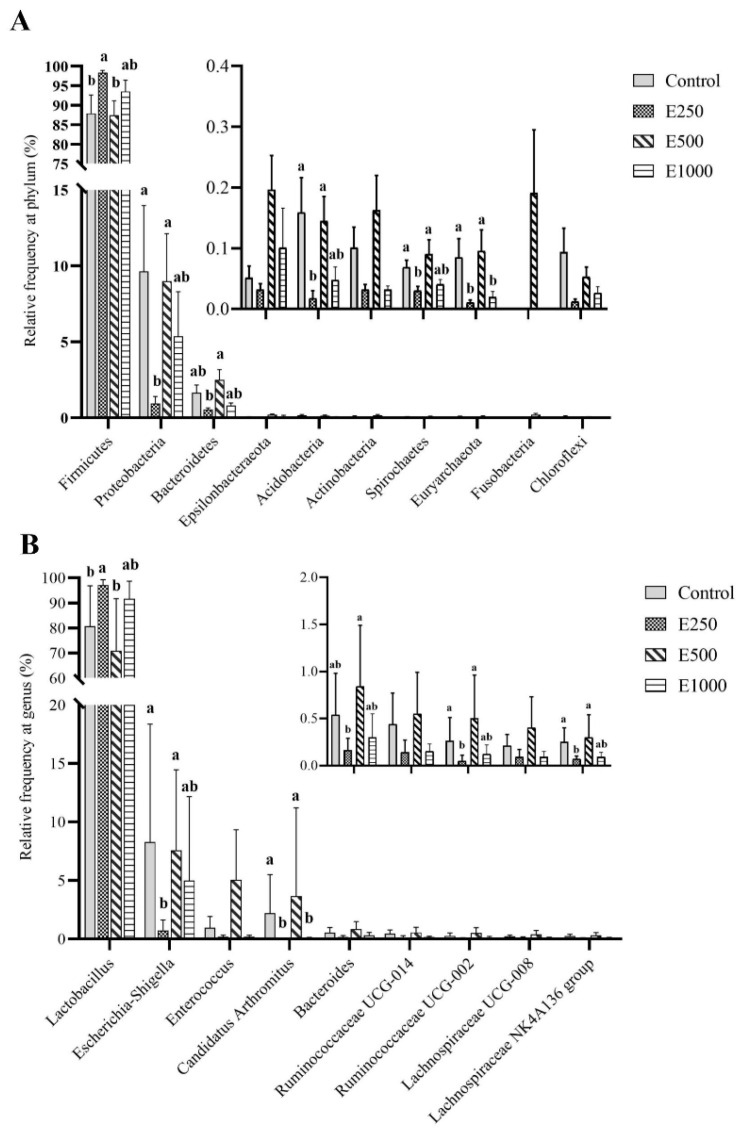
Composition of ileal microbiota at phylum (**A**) and genus (**B**) levels among different treatments. Ileal digesta samples were collected in the broilers at 42 days of age after feeding the basal diet supplemented with 0 (Control), 250 (E250), 500 (E500), or 1000 (E1000) U glucose oxidase/kg diet for 42 days. ^a,b^ Means (*n* = 6) within a column lacking a common lowercase differ (*p* < 0.05).

**Table 1 animals-11-02909-t001:** Composition and nutrient levels of the basal diets (as fed-basis).

Items	Starter Phase d 1 to 21	Grower Phase d 22 to 42
Ingredient, g/kg		
Ground corn	547.4	597.0
Soybean meal (43.0% CP)	376.2	323.9
Soybean oil	36.0	43.5
Ground limestone	11.8	11.2
Dicalcium phosphate	17.5	15.2
DL-Methionine (98.5%)	2.6	1.6
L-Lysine hydrochloride	0.3	—
Sodium chloride	4.0	4.0
Choline chloride	1.6	1.0
Vitamins premix ^1^	0.3	0.3
Minerals premix ^2^	2.0	2.0
Bentonite	0.3	0.3
Nutrient level ^3^, g/kg		
ME (MJ/kg)	12.39	12.84
Crude protein	213.0	191.3
Calcium	10.32	9.21
Nonphytate phosphorus	4.52	4.12
Lysine	12.43	11.21
Methionine	6.04	4.58
Arginine	14.43	13.55
Histidine	5.32	5.09
Isoleucine	8.36	7.63
Leucine	17.43	15.65
Threonine	9.47	8.63
Phenylalanine	10.78	10.06
Tyrosine	7.98	7.52
Cystine	3.57	3.42
Valine	9.86	9.31

^1^ For the starter diet, provided per kilogram of diet: vitamin A (all-*trans* retinol acetate), 10,000 IU; cholecalciferol, 3000 IU; vitamin E (all-*rac*-α-tocopherol acetate), 20 IU; vitamin K (menadione sodium bisulfate), 2.0 mg; thiamin (thiamin mononitrate), 1.6 mg; riboflavin, 6.0 mg; vitamin B_6_, 3.0 mg; vitamin B_12_, 0.0014 mg; pantothenate, 20 mg; niacin, 30 mg; folic acid 0.8 mg; biotin, 0.12 mg. For the grower diet, provided per kilogram of diet: vitamin A (all-*trans* retinol acetate), 10,000 IU; cholecalciferol, 3900 IU; vitamin E (all-*rac*-α-tocopherol acetate), 30 IU; vitamin K (menadione sodium bisulfate), 3.0 mg; thiamin (thiamin mononitrate), 2.4 mg; riboflavin, 9.0 mg; vitamin B_6_, 4.5 mg; vitamin B_12_, 0.021 mg; pantothenate, 30 mg; niacin, 45 mg; folic acid 1.2 mg; biotin, 0.18 mg. ^2^ For the starter diet, provided per kilogram of diet: Cu (CuSO_4_·5H_2_O), 8 mg; Zn (ZnSO_4_·7H_2_O), 100 mg; Fe (FeSO_4_·H_2_O), 60 mg; Mn (MnSO_4_·H_2_O), 100 mg; I (KI), 0.35 mg Se (Na_2_SeO_3_) 0.30 mg. For the grower diet, provided per kilogram of diet: Cu (CuSO_4_·5H_2_O), 8 mg; Zn (ZnSO_4_·7H_2_O), 80 mg; Fe (FeSO_4_·H_2_O), 40 mg; Mn (MnSO_4_·H_2_O), 80 mg; I (KI), 0.35 mg Se (Na_2_SeO_3_) 0.30 mg. ^3^ Determined by analyses except for ME.

**Table 2 animals-11-02909-t002:** Effect of glucose oxidase addition on the performance of broilers.

Treatments ^1^	BW ^2^ (g)			DBWG ^2^ (g)			DFI ^2^ (g)			FCR ^2^ (g:g)			Mortality (%)	*N*
	d 1	d 21	d 42	d 1–21	d 22–42	d 1–42	d 1–42	d 22–42	d 1–42	d 1–21	d 22–42	d 1–42	d 1–42	
Control	42.84	1085	2601	49.67	72.43 ^b^	60.90	62.19	151.00	106.7	1.254	2.049	1.714	1.89	10
E250	42.94	1030	2802	47.00	84.33 ^a^	65.69	60.57	165.29	112.9	1.289	1.962	1.685	1.32	10
E500	42.89	1059	2684	48.38	77.67 ^a,b^	62.90	62.52	158.67	110.6	1.292	2.050	1.721	1.42	10
E1000	42.94	1056	2743	48.29	80.33 ^a,b^	64.29	61.86	159.67	110.3	1.282	1.990	1.629	1.53	10
SEM	0.06	14.21	60.54	0.67	2.31	1.28	0.95	3.53	2.02	0.013	0.032	0.017	0.56	
*p*-value	0.549	0.163	0.328	0.158	0.018	0.237	0.506	0.360	0.207	0.139	0.153	0.463	0.532	

^a,b^ Means within a column lacking a common lowercase differ (*p* < 0.05). ^1^ Control, E250, E500, and E1000 represented the basal diet supplemented with 0, 250, 500, and 1000 U glucose oxidase/kg diet, respectively. ^2^ BW = bodyweight; DBWG = daily bodyweight gain; DFI = daily feed intake; FCR = feed consumption: bodyweight gain.

**Table 3 animals-11-02909-t003:** Effect of glucose oxidase addition on the apparent ileal digestibility of nutrients in the grower broilers (%).

Treatments ^1^	Control	E250	E500	E1000	SEM	*p*-Value
Dry matter	73.48	74.02	72.71	71.98	2.31	0.436
Energy	73.94	76.46	75.80	78.31	1.78	0.259
Crude protein (N × 6.25)	64.96	72.92	70.32	71.73	2.81	0.074
Indispensable amino acids						
Arginine	72.14 ^b^	78.85 ^a^	78.71 ^a^	80.86 ^a^	2.39	0.041
Histidine	74.41	77.38	76.9	77.72	2.23	0.285
Isoleucine	64.23 ^b^	73.87 ^a^	73.88 ^a^	74.02 ^a^	2.52	0.035
Leucine	77.92	75.86	75.28	75.87	2.24	0.765
Lysine	63.01 ^b^	72.10 ^a^	73.29 ^a^	76.34 ^a^	2.62	0.011
Methionine	57.90 ^b^	71.77 ^a^	72.43 ^a^	69.79 ^a^	2.62	0.012
Phenylalanine	79.97	76.93	76.29	77.80	2.11	0.479
Threonine	54.98 ^b^	65.17 ^a^	67.08 ^a^	69.56 ^a^	2.98	0.014
Valine	64.06	71.43	69.84	74.10	2.89	0.113
Dispensable amino acids						
Alanine	64.87	72.02	72.90	73.91	2.83	0.112
Aspartic acid	68.05	73.36	73.15	75.81	2.40	0.260
Glutamic acid	72.58	78.48	78.20	79.25	2.05	0.108
Glycine	69.50	67.24	69.22	72.11	2.67	0.633
Proline	72.26	77.32	76.51	79.09	2.10	0.192
Cysteine	55.25 ^b^	65.25 ^a^	67.32 ^a^	67.78 ^a^	2.74	0.025
Serine	63.30 ^b^	71.36 ^a^	71.80 ^a^	74.53 ^a^	2.62	0.049
Tyrosine	63.12 ^b^	73.37 ^a^	72.49 ^a^	75.21 ^a^	2.88	0.023
Total amino acids	69.14	76.65	72.20	74.44	2.40	0.151
*N*	10	10	10	10		

^a,b^ Means within a column lacking a common lowercase differ (*p* < 0.05). ^1^ Control, E250, E500, and E1000 represented the basal diet supplemented with 0, 250, 500, and 1000 U glucose oxidase/kg diet, respectively.

**Table 4 animals-11-02909-t004:** Effect of glucose oxidase addition on the intestinal morphology of broilers at 42 days of age.

Treatments ^1^	Control	E250	E500	E1000	SEM	*p*-Value
Duodenum						
Villus height (μm)	1527	1605	1596	1513	81.63	0.772
Crypt depth (μm)	213.1	247.6	207.3	231.1	17.44	0.371
Villus height/crypt depth	6.85	6.52	7.69	6.83	0.55	0.516
sIgA ^2^ (mg/g protein)	34.56	36.32	37.65	38.53	3.42	0.564
Jejunum						
Villus height (μm)	1233 ^b^	1507 ^a^	1191 ^b^	1200 ^b^	105.6	0.042
Crypt depth (μm)	265.2	271.9	210.4	240.4	27.61	0.541
Villus height/crypt depth	4.43	5.58	5.44	5.00	0.47	0.220
sIgA (mg/g protein)	63.41	66.53	67.86	62.19	2.95	0.633
Ileum						
Villus height (μm)	881.9 ^b^	1019 ^a^	1053 ^a^	970.2 ^ab^	48.59	<0.001
Crypt depth (μm)	170.0	143.1	138.1	180.7	22.04	0.456
Villus height/crypt depth	5.93	5.83	6.51	5.16	0.69	0.318
sIgA (mg/g protein)	62.43 ^b^	74.52 ^a^	65.42 ^b^	68.53 ^ab^	2.84	0.046
*N*	10	10	10	10		

^a,b^ Means within a row lacking a common lowercase differ (*p* < 0.05). ^1^ Control, E250, E500, and E1000 represented the basal diet supplemented with 0, 250, 500, and 1000 U glucose oxidase/kg diet, respectively; ^2^ sIgA = secreted immunoglobulin A.

**Table 5 animals-11-02909-t005:** Effect of glucose oxidase addition on the pH of intestinal digesta of broilers at 42 days of age.

Treatments ^1^	Control	E250	E500	E1000	SEM	*p*-Value
Duodenum	5.99	5.90	6.13	6.25	0.13	0.405
Jejunum	6.19 ^a^	6.25 ^a^	5.85 ^b^	6.21 ^a^	0.08	0.042
Ileum	6.18 ^a^	5.46 ^b^	6.33 ^a^	5.34 ^b^	0.17	0.024
*N*	10	10	10	10		

^a,b^ Means within a row lacking a common lowercase differ (*p* < 0.05). ^1^ Control, E250, E500, and E1000 represented the basal diet supplemented with 0, 250, 500, and 1000 U glucose oxidase/kg diet, respectively.

**Table 6 animals-11-02909-t006:** Effect of glucose oxidase addition on the concentrations of short chain fatty acids (SCFAs) in the ileal digesta of broilers at 42 days of age.

Treatment ^1^	Acetic Acid (nmol/g)	Propionic Acid (nmol/g)	Butyric Acid (pmol/g)	Iso-Valeric Acid (pmol/g)	Valeric Acid (pmol/g)	Total SCFAs (nmol/g)	*N*
Control	1.089 ^b^	0.428	41.82 ^b^	15.77	31.10	1.108 ^b^	10
E250	2.187 ^a^	0.522	78.73 ^a^	18.33	29.64	2.612 ^a^	10
E500	0.921 ^b^	0.418	34.28 ^b^	21.05	38.61	1.236 ^b^	10
E1000	0.819 ^b^	0.496	44.53 ^b^	14.43	23.06	0.739 ^b^	10
SEM	0.235	0.040	11.99	1.91	7.46	0.385	
*p*-value	0.002	0.236	0.043	0.106	0.575	0.017	

^a,b^ Means within a column lacking a common lowercase differ (*p* < 0.05). ^1^ Control, E250, E500, and E1000 represented the basal diet supplemented with 0, 250, 500, and 1000 U glucose oxidase/kg diet, respectively.

**Table 7 animals-11-02909-t007:** The effect of glucose oxidase addition on the alpha diversity indices of ileal microbiota in broilers.

Treatment ^1^	Observed Species	Chao	ACE	Shannon	Simpson	*N*
Control	446 ^ab^	717 ^a^	803 ^a^	2.46	0.76	6
E250	180 ^c^	273 ^b^	296 ^b^	1.98	0.73	6
E500	453 ^a^	705 ^a^	771 ^a^	2.61	0.75	6
E1000	226 ^bc^	382 ^ab^	414 ^ab^	1.75	0.62	6
SEM	69	105.5	119.7	0.34	0.06	
*p*-value	0.032	0.014	0.006	0.536	0.364	

^a–c^ Means within a column lacking a common lowercase differ (*p* < 0.05). ^1^ Control, E250, E500, and E1000 represented the basal diet supplemented with 0, 250, 500, and 1000 U glucose oxidase/kg diet, respectively.

## Data Availability

The data presented in this study are available on request from the corresponding author.

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
