# Peer review of "Effects of Dietary Glucose Oxidase Supplementation on the Performance, Apparent Ileal Amino Acids Digestibility, and Ileal Microbiota of Broiler Chickens"

_animals, 2021, doi:10.3390/ani11102909_

Round 1

Reviewer 1 Report

Page 8 Line 1 change “p = 0.03” as “p = 0.042”

Page 8 Line 5 change “p = 0.03” as “p <0.001”

Page 8  lines 7-9     change “The E250 treatment increased the concentration of sIgA in the ileum when compared to the control, and there was no significant difference between the control, E500, and E1000 groups” as “The E250 treatment increased the concentration of sIgA in the ileum when compared to the control, and E500 groups”

Page 13 line 2 after  “including E. coli [42].”   also adds another reference (Sigolo, S.; Milis, C.; Dousti, M.;  Jahandideh, E.; Jalali, A.; Mirzaei, N.; Rasouli, B.; Seidavi, A.; Gallo, A.; Ferronato, G.; Prandini, A.  Effects of different plant extracts at various dietary levels on growth performance, carcass traits, blood serum parameters, immune response and ileal microflora of Ross broiler chickens. Ital. J. Anim. Sci. 2021, 20, 359-371.)

Author Response

Reviewer 1:

Page 8 Line 1 change “p = 0.03” as “p = 0.042”

Response: Thanks. We revised in page 8, line 300.

Page 8 Line 5 change “p = 0.03” as “p <0.001”

Response: Thanks. We revised in page 8, line 304.

Page 8 lines 7-9 change “The E250 treatment increased the concentration of sIgA in the ileum when compared to the control, and there was no significant difference between the control, E500, and E1000 groups” as “The E250 treatment increased the concentration of sIgA in the ileum when compared to the control, and E500 groups”

Response: Thanks. We revised according to your suggestion in page 8, lines 307-308.

Page 13 line 2 after “including E. coli [42].” also adds another reference (Sigolo, S.; Milis, C.; Dousti, M.; Jahandideh, E.; Jalali, A.; Mirzaei, N.; Rasouli, B.; Seidavi, A.; Gallo, A.; Ferronato, G.; Prandini, A.  Effects of different plant extracts at various dietary levels on growth performance, carcass traits, blood serum parameters, immune response and ileal microflora of Ross broiler chickens. Ital. J. Anim. Sci. 2021, 20, 359-371.)

Thank you so much. We added this reference in cited literature 43 in page 14, line 451.

Reviewer 2 Report

General comments: The experiments did not have a “positive” treatment with antibiotics (AGP). So all comments and discussion in relation to AGP must be deleted. Anyway, GOD and AGP have probably different modes of action in the digestive tract.

English: The text is full of small errors (e.g. singular / plural, present / past, etc.).

In several tables you use to many digits after the comma point.

Explain abbreviations in tables, and if necessary, explain them (e.g. Tab. 7).

Use superscripts (e.g. Tab. 1 & 2)

Details in text cannot me mentioned. There is no line numbering.

Tab1:     Declare, what values were analyzed or calculated. Give analyzed values for all experimental diets, including GOD!

2.3.        Feed intake was measured and not calculated. Give DBWG and DFI values per day (Tab. 2).

2.4.        Sample Collection for intestinal wall analysis and morphology.

2.6.        How were AA analyzed (HPLC or AA analyzer?

Tab. 2:  Add values for mortality etc. DFI and DBWG (see earlier).

Tab. 3    Add value for digestibility of organic matter and energy.

3.6. Biodiversity etc. Put your comments in a better context, explain abbreviations, etc. What means 553, 27, 390 and 85 OUT’s?

  1. Discussion: What is Beijing You chicken?

Author Response

Reviewer 2:

General comments: The experiments did not have a “positive” treatment with antibiotics (AGP). So all comments and discussion in relation to AGP must be deleted. Anyway, GOD and AGP have probably different modes of action in the digestive tract.

Response: Thank you so much. We revised the description related to antibiotics in simple summary and introduction parts in line16-17, and 47-50.

English: The text is full of small errors (e.g. singular / plural, present / past, etc.).

Response: Thanks. We revised the whole manuscript, especially these grammar errors. We also let Wei Zhang from the University of Maryland School of Medicine help us to review the whole manuscript.

In several tables you use to many digits after the comma point.

Response: Thanks. We revised according to your suggestion in the tables.

Explain abbreviations in tables, and if necessary, explain them (e.g. Tab. 7).

Response: Thanks. We revised according to your suggestion in table 7.

Use superscripts (e.g. Tab. 1 & 2)

Response: Thanks. We revised according to your suggestion in tables 1 and 2.

Details in text cannot me mentioned. There is no line numbering.

Response: Thanks. We added the line number in the whole manuscript.

Tab1: Declare, what values were analyzed or calculated. Give analyzed values for all experimental diets, including GOD!

Response: Thank you so much for your suggestion. We have provided the analyzed nutrient level in Table 1. Because the different diets were formulated from the same basal diet, we therefore only determine the nutrients level in the basal diet. For the GOD, we have determined the activity of GOD before the GOD was added in the different treatment diet (Material and method). However, we did not measure the activity of GOD after the different diet was formulated, because there was not a feasible method, to our knowledge, to determine the GOD in the mixed feed.

2.3. Feed intake was measured and not calculated. Give DBWG and DFI values per day (Tab. 2).

Response: Thanks. We revised according to your suggestion in table 2.

2.4. Sample Collection for intestinal wall analysis and morphology.

Response: Thanks. We revised the description according to your suggestion

2.6. How were AA analyzed (HPLC or AA analyzer)?

Response: Thanks. We measured the concentrations of amino acids using AA analyzer. We revised the description in page 4, lines 168-169.

Tab. 2: Add values for mortality etc. DFI and DBWG (see earlier).

Response: Thanks. We added the data of mortality, DFI, and DBWG in Table 2.

Tab. 3 Add value for digestibility of organic matter and energy.

Response: Thanks for your meaningful comment. We added the data of the digestibility of dry matter and energy in Table 3. But we did not measure the digestibility of organic matter, and we will consider it in the further research.

3.6. Biodiversity etc. Put your comments in a better context, explain abbreviations, etc. What means 553, 27, 390 and 85 OUT’s?

Response: Thanks. We revised the “OUT” into “OTU”, and added the full name of OTU in the first appearance (line 339). We also added some comments in the discussion in page 14, lines 445-447.

4. Discussion: What is Beijing You chicken?

Response: Thanks. Beijing You chicken is a slow growing local broilers in China. We revised the description as “slow growing broiler chickens”.

Reviewer 3 Report

Please find attached comments to the Manuscript.

Overall the Manuscript needs a Grammar review. A native English speaker would be of great help to address English issues. 

Round 2

Reviewer 2 Report

Give n values for statistical analysis in Tables 2 - 7

Author Response

Replies to the reviewer’s comments

Reviewer 2

English language and style are fine/minor spell check required.

Response: We have carefully revised the manuscript according to the reviewer's comments.

Give n values for statistical analysis in Tables 2 – 7

Response: We provide the n values in Tables 2-7.

Reviewer 3 Report

Good work addressing comments

Author Response

Replies to the reviewer’s comments

Reviewer 3

English language and style are fine/minor spell check required.

Response: We have carefully revised the manuscript according to the reviewer's comments.

Good work addressing comments.

Response: Thanks reviewer for good comments and hard work.
